# Morphological, Molecular, and Alkaloid Gene Profiling of *Epichloë* Endophytes in *Elymus cylindricus* and *Elymus tangutorum* from China

**DOI:** 10.3390/microorganisms13102275

**Published:** 2025-09-28

**Authors:** Taixiang Chen, Wencong Liu, Kai Huang, Gensheng Bao, Chunjie Li

**Affiliations:** 1State Key Laboratory of Herbage Improvement and Grassland Agro-Ecosytems, College of Pastoral Agriculture Science and Technology, Lanzhou University, Lanzhou 730020, China; chentx@lzu.edu.cn (T.C.); 220220902320@lzu.edu.cn (W.L.); 2Dingxi Academy of Agricultural Sciences, Dingxi 743000, China; dxnkyhuangkai@163.com; 3State Key Laboratory of Plateau Ecology and Agriculture, Qinghai University, Xining 810016, China; baogensheng2008@hotmail.com

**Keywords:** phylogenetics, mating type genes, alkaloid diversity

## Abstract

*Epichloë* endophytes are mutualistic associates with grasses, conferring host plants with enhanced competitiveness, improved stress tolerance, and increased ecological dominance. *Epichloë* can produce any of several classes of bioactive alkaloids, of which indole-diterpenes and ergot alkaloids are toxic to invertebrate and mammalian herbivores; peramine acts as an insect-feeding deterrent; and loline alkaloids possess potent insecticidal activity. Here, it was characterized as *Epichloë* endophytes inhibiting the *Elymus* species, *El. cylindricus*, and *El. tangutorum* from the Qinghai–Tibet Plateau, China. Based on the results of four types of alkaloid synthesis genes, the 30 isolates were divided into five types; they exhibited distinct alkaloid synthesis capabilities, highlighting intraspecific diversity within *E. bromicola* regarding its alkaloid-producing potential. Considering the toxicity of these isolates to the safety of herbivorous livestock, the above five types of isolates can be divided into two categories. Category I includes five animal-safe strains of type V, which do not produce alkaloids. Category II includes the remaining 25 strains, which could produce indole-diterpene (paspaline) and/or ergot alkaloids (chanoclavine I, D-lysergic acid, ergovaline) that are toxic to herbivorous livestock. Morphology and phylogenetic analysis confirmed all 30 isolates were *Epichloë bromicola*; mating type gene detection shows that all belonged to mating type A. Overall, this study has laid a solid foundation for the scientific and rational utilization of *Epichloë* endophyte resources. Furthermore, the presence of ergovaline in *El. cylindricus* and *El. tangutorum* poses a potential concern for livestock managers who conduct grazing.

## 1. Introduction

The *Epichloë* genus (Clavicipitaceae, Ascomycota) are a class of fungal endophytes that typically establish mutualistic symbiotic relationships with cool-season grasses, and they can complete all or part of their life without causing obvious disease symptoms in the host [1,2]. Many cool-season grasses often have symbiotic relationships with *Epichloë*. The Poaceae family is the only recorded host of *Epichloë* endophytes. Up to now, *Epichloë* endophytes have been found in more than 300 species of grasses in 80 genera worldwide [3,4,5,6,7]. These endophytes are found in the intercellular spaces of aerial tissues of hosts [3]. Species of the *Epichloë* genus that sexually reproduce are capable of forming fruiting bodies known as stromata, which envelop the developing inflorescences and inhibit seed formation—resulting in “choke” diseases in host plants, and consequently facilitating horizontal transmission. However, some can colonize the host in an asymptomatic state and spread vertically; the asexual reproductive form previously classified as a species of *Neotyphodium* spp. is still benign and grows into the developing ovules, spreading through the seeds of the mother plant lineage. Some *Epichloë* species can be facultative in their dispersal, using both seeds and ascospores for transmission [8,9]. However, vertical transmission remains the main mode of dispersal for *Epichloë* species [10]. To date, based on morphological, host specificity and phylogenetic analyses, more than 50 species have been described. The phylogenetic analysis of the intron-rich sequences from the translation elongation factor 1-a gene (*tefA*) and the β-tubulin gene (*tubB*) has been employed to substantiate the evolutionary relationship among *Epichloë* species.

*Epichloë* form reciprocal symbionts with host plants, host plants provide the living environment and nutrients for *Epichloë*, and in return, some *Epichloë* are known to provide fitness benefits to their host plants including drought, salinity, low temperature, flooding, heavy metals tolerance, disease resistance, enhance competitiveness, improve survival ability [2,11], and protection against pests and mammalian herbivory through the production of alkaloid compound [12,13,14]. The interaction between symbionts is mainly played by secondary metabolites [15]. Hence, these secondary metabolites play an important role in the symbiotic relationship between host plants and *Epichloë* endophyte, of which alkaloids account for nearly half [16]. As early as 1997, after Bacon et al. found that alkaloids of *Epichloë* could cause livestock poisoning, researchers paid extensive attention to the alkaloids and thus conducted a series of studies [17]. Common alkaloids can be divided into four main groups: ergot alkaloids (such as ergovaline), indole-diterpenes (such as lolitrems B), bispyrrolidines (loline), and pyrrolipyrazines (peramine). Peramine is an insect-feeding deterrent, whereas loline has potent insecticidal activity. Indole-diterpenes (lolitrems B) and ergot are most commonly documented for being toxic to vertebrates, causing ryegrass staggers and fescue toxicosis, respectively [18,19]. Based on the completion of whole genome sequencing, four major classes of alkaloid-synthesis-related genes have been elucidated. The *ppzA* (formerly known as *perA*) gene located at the PER locus is essential for the biosynthesis of peramine [20,21]. However, the *loci* required for the production of loline, indole-diterpenoids, and ergot alkaloids are all complex gene clusters containing large stretches of related repetitive elements [22,23,24,25]. The diversity of alkaloids produced by metabolic pathways depends on the genes present in the genome, and the presence of functional domains in biosynthetic enzymes. Therefore, the presence or absence of certain genes and the sequences of the proteins they encode can be used to predict the ability of *Epichloë* to produce alkaloids or related intermediates [26,27,28]. This prediction represents a cost-effective and rapid method for screening the individuals of large *Epichloë*–grass symbiotic populations likely to be useful in forage agriculture [28]. This provides a theoretical basis for fully utilizing the alkaloid characteristics of *Epichloë* endophytes [29].

The diversity of alkaloids has promoted the development of novel *Epichloë*, which do not produce toxic alkaloids harmful to vertebrates. These novel *Epichloë* have been integrated into the biological breeding program to inoculate *Epichloë*-free plants with isolates that only synthesize insect-resistant alkaloids, without synthesizing alkaloids that cause toxicity to livestock, thereby establishing a new animal-safe *Epichloë*–host symbiotic association [30,31]. The goal is to accelerate germplasm innovation by leveraging multiple stress-resistance enhancement traits of *Epichloë* for their host and their vertical transmission characteristics, while with nil or much reduced (acceptable and manageable) adverse impacts on livestock [32]. This method aims to transfer the beneficial traits conferred by *Epichloë* from the natural host to the new host. However, the success of this inoculation process depends on various factors, including the inoculation technique, the *Epichloë* isolate, host genotype, and host specificity [33]. Consequently, the identification and exploration of *Epichloë* isolates’ characteristics is crucial for advancing this research. Through this process, a number of novel *Epichloë* isolates have been delivered and are now commercially used in USA and New Zealand, such as AR1^TM^, AR37^TM^, Endo5^TM^, and NEA^TM^ endophyte for ryegrass, E34^TM^, AR542^TM^, MaxQ^TM^, and MaxP^TM^ for tall fescue. Effective delivery of novel *Epichloë*-infected cultivars requires care with management of seed and quality control systems, the monitoring of *Epichloë* viability is required through the distribution chain, and the seed must be stored at relative low humidity and low temperature until ready to be sown.

*Elymus* spp. plants are the largest genus in the Poaceae family. Among these plants, *Elymus cylindricus* and *Elymus tangutorum* are excellent herbage in western China; they possess remarkable nutritional value and palatable taste, making them an important choice for animal feed. Their root systems are well-developed, enabling them to have a strong soil and water conservation ability and they have become one of the important native grasses for the restoration and reconstruction of degraded grassland on the Qinghai–Tibet Plateau [34,35]. *El. tangutorum* is known to harbor *Epichloë* endophyte [35,36], and many other *Elymus* species have been reported to be associated with *Epichloë elymi*, *Epichloë bromicola*, and *Epichloë canadensis*. The genes responsible for the biosynthesis of ergot alkaloids, indole-diterpenes, bispyrrolidines (loline), and pyrrolipyrazines (peramine) in *Epichloë* endophyte symbiotic with *El. cylindricus* and *El. tangutorum* from the Qinghai–Tibet Plateau have not yet been characterized. In this study, *El. cylindricus* and *El. tangutorum* were collected from four different locations in the Qinghai–Tibet Plateau, and 30 *Epichloë* strains were isolated. Morphology and phylogenetic methods were used to characterize the asexual *Epichloë* of *El. cylindricus* and *El. tangutorum*. Further, alkaloid production potential was conducted based on PCR profiling of genes required for the four kinds of alkaloid biosynthetic pathways. The results contribute to the knowledge about grass–*Epichloë* symbioses and lay a solid foundation for further utilization of *Epichloë* in germplasm innovation.

## 2. Materials and Methods

### 2.1. Collection and Preservation of Plant Materials and Endophytes

The *El. tangutorm* and *El. cylindrical* specimens were collected between August and September 2023, with mature seeds obtained from Haixi, Hainan, Guoluo, and Yushu prefectures in Qinghai Province, China (Table 1). The total number of plants sampled at each location is presented in Table 1. To ensure the independence and representativeness of the samples, only one tiller was collected per plant. Leaf sheaths of samples were microscopically examined using the aniline blue staining method described by Li (2008); healthy stem tissue was ruptured gently, stained with 0.05% aniline blue solution, and then viewed under an optical microscope [37]. Half of the seeds from endophyte-infected (E+) plants were used for *Epichloë* endophyte isolation. The rest of the plants were stored at 4 °C to maintain the viability of the *Epichloë* endophyte and seeds.

### 2.2. Isolation of Endophytes and Morphological Examination

Infected seeds were surface sterilized with 70% ethanol for 3 min, placed into 5% sodium hypochlorite for another 3 min, then rinsed three consecutive times with sterile water, and left on sterile filter paper to dry. A group of 3–5 seeds was placed on each PDA plate supplemented with 100 μg/mL ampicillin and 50 μg/mL streptomycin [38]. The plates were sealed and incubated at 22 °C in the dark, with daily checks for contamination. After the mycelium grew out, it was purified three times to obtain the pure culture isolate. Mycelial disks (0.4 cm diameter) were placed in the center of the PDA medium and incubated at 22 °C in the dark for 32 days. Colony morphology was observed and photographed [39]. Growth rate experiments were conducted by placing mycelial disks on a PDA medium with six replicates per strain and incubating at 25 °C in the dark, colony diameter was measured weekly over a 56 d period using the cross-crossing method, and the daily growth rate for each strain was calculated [37,40,41]. The results presented are expressed as the mean ± standard error (Table 2). A drop of sterile water was placed in the center of the slide, a cover slide with mycelium was placed on it, and the morphology of conidia and conidiogenous cells was observed and photographed using a fluorescence microscope (Olympus BX63, Olympus, Tokyo, Japan). Images were captured with an Olympus BX51 camera supported by Cellsens Entry 1.8 software (Olympus Corp., Tokyo, Japan). Images were used to measure the length and width of conidia (30 conidia were measured for each isolate) and the length of conidiogenous cells (20 conidiogenous cells were measured for each isolate). Results are presented as mean ± standard error (Table 2). The morphologies of the isolates were compared with those of other *Epichloë* species, including species isolated from *Elymus* spp.

### 2.3. DNA Extraction, PCR Analyses, and Phylogenetic Analysis

Total DNA was extracted from *Epichloë* endophytes. After each strain was purified and cultured on PDA medium for four weeks, mycelia on the surface of the medium were scraped using a sterile slide, and the total DNA of *Epichloë* endophyte was extracted using a fungal kit with the instructions of the manufacturer (Omega, Beijing, China). Following Moon et al. (2002) [52], PCR amplification was carried out with two sets of housekeeping genes: *tefA*, which is an elongation factor gene, and *tubB*, which codes for tubulin B. For *tubB*, the primers were tub2-exon 4u-2 and tub2-exon 1d-1 [52]. For *tefA*, the primers were tef1-exon 5u-1 and tef1-exon 1d-1 [53]. The 25 µL PCR amplification mixture consisted of 12.5 µL of 2 × San Taq Fast PCR Mix, 1 µL (10 µM) of each of the target-specific primer, 9.5 µL of ddH_2_O, and 1 µL of DNA at a concentration of 40 ng/µL. The PCR cycling conditions used were one cycle of initial denaturation at 94 °C for 5 min, then 34 cycles of denaturation at 94 °C for 30 s, annealing at 52 °C (for *tubB*) or 55 °C (for *tefA*) for 30 s, extension at 72 °C for 1 min, and finally one cycle for synthesis at 72 °C for 10 min [54,55]. The amplified products were analyzed by electrophoresis using 1.5% agarose gel in 1 × TAE buffer. DNA fragments were stained with Gold View (Solarbio Corp., Beijing, China) and viewed by UV transillumination. The products were sent to Shanghai BioEngineering Company (Shanghai, China) for sequencing. The sequences were submitted to GenBank (Accession numbers: *tubB*: PQ157737~PQ157766; *tefA*: PQ119943~PQ119947 and PQ157712~PQ157736). The sequences were compared with known *Epichloë* spp. using the online tool MAFFT (v. 7.526) (https://mafft.cbrc.jp/alignment/server/index.html (accessed on 13 September 2024)) [56]. Bioedit v. 7.20 software checked and optimized the comparison results [57]. IQtree (v. 2.3.6) software was used to select the most suitable tree-building model and construct the maximum likelihood (ML) phylogenetic tree [58]. Use Interactive Tree of Life (ITOL) (v. 7.2) (https://itol.embl.de/ (accessed on 15 September 2024)) to view and beautify evolutionary trees [59]. The housekeeping genes *tubB* and *tefA* were used to construct a phylogenetic tree. *tefA* sequence of *Claviceps purpurea* (GenBank accession number AF276508) and *tubB* sequence of *C. purpurea* (GenBank accession number AF062646) were used as the outgroup of the tree [50,52].

### 2.4. Alkaloid Gene Detection

Based on PCR amplification, the endophytes were tested for the presence of mating type and alkaloid synthesis genes using specific primers, as described previously [51,60]. About 46 genes related to *Epichloë* endophyte were detected. In total, 2 mating type genes, 8 segments of the *ppzA* gene, which is involved in peramine biosynthesis, 14 *EAS* cluster genes involved in ergot alkaloid biosynthesis, 11 *IDT/LTM* cluster genes required for indole-diterpene production, and 11 *LOL* cluster genes required for loline biosynthesis were identified. Using the total DNA of *Epichloë* endophyte, specific primers for each of the 46 genes were selected for PCR amplification. The PCR cycling parameters were set to 94 °C for 1 min, followed by 30 cycles of 94 °C for 15 s, 56 °C for 30 s, 72 °C for 1 min, and then final annealing at 72 °C for 10 min [51,60]. This cycle was used for all the primers. The PCR products were detected by 1.5% agarose gel electrophoresis and viewed by UV transillumination. *E. inebrians* strain E818 was used as a positive control for *EAS* gene cluster genes [10], and *E. festucae* var. *lolii* strain AR1 was used as a positive control for *ppzA* and some genes at the *IDT* locus [24,61]. The endophytic fungus *Epichloë* sp. FS001 isolated from *Festuca sinensis* was also used as a positive control for the *ppzA* and *IDT* gene cluster genes [62].

## 3. Results

### 3.1. Isolation of Endophytes

Table 1 shows that *El. cylindricus* and *El. tangutorum* were collected from Qinghai Province, China. There were 514 individual plants of the two grasses, 83 of which harbored endophytic fungi of the *Epichloë* genus, resulting in a total *Epichloë* infection rate of 16.15% (Table 1). For *El. cylindricus*, a total of 372 individual plants were collected, of which 61 were found to harbor *Epichloë* endophyte, resulting in a total infection rate of 16.4%. The *Epichloë* infection rate among plants from different regions ranged from 5.88% to 60%, with the highest rate found in the Haixi prefecture. For *El. tangutorum*, a total of 142 individual plants were collected from Haixi, Hainan, and Yushu prefectures, of which only 22 were found to carry *Epichloë* endophyte, yielding a total infection rate of 15.5%. The *Epichloë* infection rate for this species ranged between 4.34% and 40%, with the Haixi prefecture again exhibiting the highest rate. A total of 30 *Epichloë* endophytes strains were isolated from surface-sterilized *Epichloë* endophyte-infected samples, including 17 strains from *El. cylindricus* and 13 strains from *El. tangutorum* (Table 1).

### 3.2. Morphological Examination

All 30 strains exhibited typical characteristics of *Epichloë* endophyte in the PDA medium. Due to different environmental conditions such as sampling host, altitude, latitude, and longitude, 30 strains showed some differences in colony morphology, conidia size, conidiogenous length, growth rate, and other morphological characteristics. In culture on PDA after 30d at 25 °C, the colonies’ characteristics of the strains can be summarized into two categories: in the first type, the middle of the mycelia is white, the edge is light yellow, the center has a spherical bulge, and the colony edge is regular and cotton-like at the top; in the second type, the mycelium texture is dense, the cotton is white, and the colony edge is regular (Figure 1). The reverse of colonies is light brown centrally to cream at the margin, and the color of the isolates MB6, MB8, YZ8, YZ12, and YZ14 gradually changed from dark brown in the middle to pale yellow on both sides. The mature conidia could be seen after 40 days of cultivation on PDA medium. The conidia were boat-shaped to oval-shaped and had single terminals. For 30 isolates, the length of the conidia ranged from 3.45 to 5.47 μm, the width from 1.27 to 3.37 μm, and the length of the conidiogenous cell from 9.09 to 16.55 μm. The 30-day colonies range in diameter from 1.38 to 3.67 cm, with an average growth rate ranging from 0.63 to 1.18 mm/day (Table 2). The isolates from two grasses differ in conidia size, conidiogenous cell size, and growth rate. For *El. cylindricus*, the length of the conidia ranged from 3.45 to 4.64 μm, the width from 1.27 to 2.79 μm, the length of the conidiogenous cell from 9.09 to 25.48 μm, and the growth rate from 0.63 to 1.18 mm/day. For *El. tangutorum*, the length of the conidia ranged from 3.5 to 5.47 μm, the width from 1.66 to 3.37 μm, the length of the conidiogenous cell from 9.56 to 16.55 μm, and the growth rate from 0.78 to 1.11 mm/day. Significant differences in conidia size, conidiogenous cell size, and growth rate existed between isolates from two hosts and between isolates from the same host.

The morphological characteristics of the other *E. bromicola* strains listed in Table 2 included *E. bromicola* from *Bromus ramosus*, *B. benekenii*, *B. erectum*, *Hordeum brevisubulatum*, *Hordelymus europaeus*, *H. bogdanii*, *Leymus chinensis*, *Psathyrostachys lanuginosa*, and *Roegneria kamoji*. When compared to the previously reported *E. bromicola* endophytes, the morphological features (conidia size, length of conidiogenous cells, and growth rate) of the 30 isolates examined in this study were slightly different but still fell within the normal range. However, the morphological indexes could be easily affected by culture conditions, and the *Epichloë* identity could not be accurately identified from morphological characteristics. In the morphology of *Epichloë* species in different host plants in the *Elymus* genus, at least four different taxa, *E. bromicola*, *Epichloë elymi*, *Epichloë canadensis*, and *Epichloë glyceriae*, have been identified from different *Elymus* species.

### 3.3. Phylogenetic Analysis

The phylogenetic tree obtained from the maximum likelihood analysis based on the *tubB* gene sequence showed that the 30 strains from *El. cylindricus* and *El. tangutorum* fell into a cluster with *E. bromicola* isolated from *El. tangutorum*, *El. cylindricus*, *El. repens*, and *L. chinensis*. All these strains constituted a branch with a 100 bootstrap support value, as shown in Figure 2. According to the maximum likelihood phylogenetic tree based on the *tefA* housekeeping gene sequences of *Epichloë* endophytes, 30 strains isolated from *El. cylindricus* and *El. tangutorum* clustered with *E. bromicola* isolated from *El. tangutorum*, which had a bootstrap support value of 84 (Figure 3). Maximum likelihood phylogenetic trees constructed with housekeeping gene sequences, *tefA* and *tubB*, were congruent. Based on these results, all 30 *Epichloë* endophytes from *El. cylindricus* and *El. tangutorum* were identified as *E. bromicola*.

### 3.4. Alkaloid Gene Detection

During the amplification of mating type genes, 30 isolates in this study exhibited identical amplification profiles, being positive exclusively for the *mtAC* marker, which is indicative of mating type A (Table 3). Within the eight domain structures of the peramine synthetase-encoding gene, *ppzA* (formerly known as the *perA* gene), five isolates harbored all eight domains; in contrast, three strains were negative for all eight domains, while other strains exclusively contained one or several domains among the *ppzA* gene. The *ppzA*-∆R (representing allele *ppzA*-2) refers to the *ppzA* gene from which the R-domain has been deleted, the functional implication of this deletion is the absence of the final enzymatic step required to convert diketopiperazine into peramine in the ∆R variant, resulting in the production of pyrrolopyrazine-1,4-diones instead of peramine. Previous studies have indicated that this variant retains the capacity to encode different metabolites and may confer protective effects on their host [21]. Among the strains tested, 24 were positive for *ppzA*-∆R, suggesting their inability to produce peramine. In contrast, strains YZ8, MB8, MB11, YZ11, YZ13, and MB13 were negative for *ppzA*-∆R, but likely incapable of synthesizing peramine due to the absence of key functional domains within the *ppzA* gene.

Among the 14 genes involved in ergot alkaloid synthesis, 25 out of 30 isolates were found to harbor the genes *dmaW*, *easF*, *easC*, and *easE*, indicating their potential for synthesizing chanolavine I (CC). Seven isolates, MB2, MB3, MB6, MB7, MB8, MB11, and YZ11, contained 8 of the 14 genes associated with ergot alkaloids’ biosynthesis, including *dmaW*, *easF*, *easC*, *easE*, *easD*, *easA*, *easG*, and *cloA*, suggesting their potential to produce chanolavine I (CC) and D-lysergic acid (D-LA). Furthermore, 11 out of 14 genes at the *EAS* locus were identified in 11 isolates, implying that these strains may be capable of synthesizing chanolavine I (CC), D-lysergic acid (D-LA), and ergovaline (ERV), but no ergonovine and lysergic acid, based on the current understanding of the biosynthetic pathway. However, five isolates, YZ5, YZ7, YZ8, YZ9, and YZ14, appeared to lack a functional *easF* gene, which has been shown to be required for the biosynthesis of early intermediates in the ergot alkaloid pathway, indicating that the five isolates may not produce even early intermediates in the ergot alkaloid pathway (Table 4). Of the 11 genes within the *LTM/IDT* clusters, strains YZ16, YZ17, and MB10 contained five of them, including *idtG*, *idtB*, *idtM*, *idtC*, and *idtS*, suggesting their potential to synthesize paspaline, but not terpendole, paxilline, paspalinine, and lolitrm, theoretically. Other strains appeared to lack some functional genes such as *idtG*, *idtB*, *idtM*, and *idtC*, which have been shown to be required for the biosynthesis of early intermediates in the lolitrem B pathway, probably rendering them incapable of synthesizing any type of indole-diterpene alkaloids (Table 5). Regarding the 11 genes within the genes responsible for loline alkaloid synthesis, all 30 strains contained only the *lolC* gene and theoretically had no potential to the production of 1-acetamido-pyrrolizidine (ACAP) or other *LOL* intermediates.

Phylogenetic analysis confirmed that all 30 isolates were *E*. *bromicola*; mating type gene detection shows that all 30 strains belonged to mating type A. Based on the results of four types of alkaloid synthesis genes, 30 isolates were divided into five types (Table 6). Type I included three strains (YZ16, YZ17, MB10) which would have the ability to produce paspaline (PAS), chanoclavinel (CC), D-lysergicacid (D-LC), and ergovaline (ERV). This type is also the most abundant type of alkaloid production among the isolates. Type II included nine strains (YZ1, YZ2, YZ3, YZ4, YZ6, YZ10, YZ15, MB1, MB9) which were predicted to possess the capability to synthesize chanoclavinel (CC), D-lysergicacid (D-LA), and ergovaline (ERV). Type III comprised seven strains (YZ11, MB2, MB3, MB6, MB7, MB8, MB11), which were predicted to possess the potential to synthesize chanoclavinel (CC) and D-lysergicacid (D-LA). Type IV included six strains (YZ12, YZ13, MB4, MB5, MB12, MB13), which had the potential to synthesize chanoclavinel (CC). Type V included five strains (YZ5, YZ7, YZ8, YZ9, YZ14), which had no theoretical potential to synthesize any kind of alkaloids.

## 4. Discussion

It was demonstrated that *E. bromicola* is a common seed-transmitted endophyte of *El. cylindricus* and *El. tangutorum*, which are important rangeland forage grasses in north China, and that *E. bromicola* is vertically transmitted in *El. cylindricus* and *El. tangutorum* seeds but very rarely causes the chock disease. Furthermore, variation was found in the genes for biosynthesis of peramine, ergot, and indole-diterpene alkaloids, and it was shown that some isolates may have the ability to produce complex ergot alkaloids such as ergovaline, which are known to be toxic to insects and livestock [26], while 5 isolates out of 30 were found to lack genes responsible for the biosynthesis of any alkaloids.

In previous studies, it has also been found that *E. bromicola* endophytes can infect many other host plants, for example, *H*. *brevisubulatum* [42], *L*. *chinensis* [45,63], *P*. *lanuginosa* [63], and are further found in a large number of *Elymus* spp. plants, including *El. dahuricus*, *El. excelsus*, *El. nutans*, and *El. tibeticus* [16]. In a study by Du et al. (2024), all 20 *E. bromicola* strains associated with five *Elymus* spp. from five regions in northwest China were identified as mating type A (MTA) [36]. Yi et al. (2018) conducted a study analyzing *E. bromicola* from six *Hordeum* seed accessions, all of which were classified as MTA [44]. *E. bromicola* isolates obtained from symbiotic associations with *H. brevisubulatum* were also found to belong exclusively to MTA [42]. In another study, *E. bromicola* isolates derived from *El. dahuricus* revealed that all but one of the ten isolates belonged to mating type A (MTA) [49]. Among the eight *E. bromicola* isolates from *El. kamoji*, six were classified as mating type B (MTB), while the remaining two were categorized as MTA [47]. Similarly, Chen et al. (2024) found two MTA and two MTB isolates from *P. lanuginosa* [46]. In the present study, all 30 isolates were categorized as MTA. The above studies collectively suggested that mating type diversity in *E. bromicola* is relatively low. However, the simultaneous presence of both mating type A (MTA) and mating type B (MTB) in certain *E. bromicola* populations suggests the potential existence of a sexual stage in specific cases, despite the absence of observed stromata on *El. cylindricus* and *El. tangutorum* under natural field conditions. Surveys of *Epichloë* endophyte infection in *El. dahuricus* across its natural distribution range in Xinjiang, Shanxi, and Beijing, China, have revealed highly variable infection frequencies, ranging from 0% to 100%. *E. bromicola* is the predominant, and often sole, *Epichloë* species associated with *El. dahuricus*; however, its sexual stage has not been observed in this host [49]. Similarly, the sexual reproductive stage of *E. bromicola* strains isolated from five *Elymus* species collected in various regions of northwest China remains unobserved [36]. Originally identified from *Bromus* spp. grasses, *E. bromicola* was first described as a common choke disease pathogen of *B. erectus* and as a strictly seed-transmitted endophyte in *B. benekenii* and *B. ramosus* within the grass tribe Bromeae [36]. The formation of stromata in *El. cylindricus* and *El. tangutorum* warrants further investigation.

In this study, the morphological characteristics of *Epichloë* endophyte isolated from different grass species and the same grass species often differed in the PDA medium, mainly in conidia size, conidiogenous cell size, and growth rates. This has been seen in previous studies; *Epichloë* endophytes carried by the same species of grass and the morphological characteristics of the *Epichloë* endophyte of the same species vary depending on the geographical region [44]. For example, *Epichloë* endophytes exhibiting distinct morphological characteristics were isolated from *Festuca arizonica*, a dominant perennial bunchgrass found in ponderosa pine–grassland communities in the southwestern United States [64]. In addition, three isolates from surface-sterilized seeds of *H*. *bogdanii* showed different morphological characteristics in culture [65].

The salient feature of *Epichloë* endophytes is that they provide a direct defense to their host by producing protective alkaloids that prevent the host from being consumed by herbivorous insects or herbivores [39,66]. This study comprehensively examines the *Epichloë* endophytes associated with seeds of *El. cylindricus* and *El. tangutorum*. The 30 strains exhibited distinct alkaloid biosynthetic capabilities. Alkaloid biosynthesis potential differed significantly among strains from host plants and habitats [42,49]. This phenomenon is consistent with the results of this study where 30 isolates highlight intraspecific diversity within *E. bromicola* regarding their alkaloid-producing potential; examples of this phenomenon are seen in *E. bromicola* isolated from *El. dahuricus* [49], *El. tangutorum*, *El. cylindricus*, *El. sibiricus*, *El. nutans* [36], and *P. lanuginosa* [46]. Studies on the biosynthetic pathway of alkaloids based on genomics and gene silencing technology show that the presence or absence of genes in the middle or at the end of the biosynthetic pathway greatly increases the diversity of alkaloids [26,67]. The diversity of indole-diterpenoid–alkaloid profiles is related to *IDT* cluster genes [15], which contain many transposon-derived repeats, which may promote rearrangement and partial or complete deletion of genes [26]. In this study, 27 strains lacked the potential to synthesize lolitrem B and other intermediate metabolites due to the lack of one or more genes required to synthesize the first intermediate product. The diversity of ergot alkaloid profiles is associated with changes in the organization, gene content, and gene sequence of *EAS* clusters [15]. Due to the lack of *lpsC*, *easO*, and *easP* genes, all 30 *Epichloë* endophytes were probably unable to produce ergonovine (EN) and lysergic acid α-hydroxyethylamide (LAH). Five strains lacked *easF*, *easC*, and *easE* gene fragments and did not have the potential to synthesize any ergot alkaloids. This phenomenon has been previously observed in *E. festucae*. Genome sequencing of two isolated *Epichloë* endophytes, F11 and E2368, from *E. festucae* revealed that isolate F11 was capable of synthesizing the ergot alkaloid end products ergovaline and peramine, as well as the indole-diterpene end product lolitrem B. In contrast, E2368 produced only ergot alkaloid end products [26]. This phenomenon is also found in the endophyte of *F*. *rubra* subsp. *commutate* [24,68]. In summary, there is considerable variation in alkaloids between different *Epichloë* species and even within species [68,69]. This intraspecific alkaloid diversity appeared to be influenced by both genetic polymorphism within the population and host species.

The identification of ergovaline-producing endophyte isolates YZ16, YZ17, and MB10 in *El. cylindricus* and *El. tangutorum* was unexpected since the ergovaline-producing endophyte could pose risks to livestock. No study has reported that livestock were poisoned by *El. cylindricus* and *El. tangutorum*, yet. However, the effects of ergot alkaloids are dependent on multiple factors, such as livestock genetics, stress, and dosage. Further investigations are required to evaluate whether *El. Cylindricus* and *El. tangutorum* with ergovaline-producing *E. bromicola* could affect livestock reproduction and health in current conditions. Ergovaline-producing *Epichloë coenophiala* isolates in much of more than 14 million ha of *Schedonorus arundinaceus* have caused hundreds of millions of dollars in lost productivity annually due to fescue toxicosis in United States [70,71]. The presence of ergonovine and ergine in *El. tangutorum* was documented by Shi et al.; ergovaline, terpendoles, and peramine were confirmed in *El. dahuricus* plants with *E. bromicola* in China [72]. Although the risk of toxicosis may be much less in rangelands of China due to relative high diversity of forage and other grasses, the significance of ergovaline, ergonovine, and ergine production by *Epichloë* of important forage grasses such as *El. cylindricus* and *El. tangutorum* warrants consideration for livestock that graze them and livestock management in rangelands of China. No livestock poisoning incidents were observed in the corresponding collection area of this study. This is primarily attributed to the insufficient concentration of toxic alkaloids to exceed the threshold required for livestock toxicity, or to the low biological activity of these alkaloids, which is inadequate to induce poisoning. However, further experimental investigation is necessary to assess their alkaloid production capacity in host grasses, with potential implications for applications in artificial inoculation studies.

The utilization of *Epichloë* in plant breeding has been gradually adopted due to the demonstrable benefits they contribute to the host and the ease of detecting and screening the alkaloids they produce. One strategy is to inoculate *Epichloë* species that do not synthesize harmful alkaloids into other grass species or food grain species, thereby generating novel varieties with desirable properties [73]. However, the actual results are often suboptimal, as some materials that have been successfully inoculated with *Epichloë* strains may exhibit poorer performance, such as severe stunted growth [74]. Researchers have introduced *E. bromicola* from *El. dahuricus* and *El. mutabilis* into *Triticum aestivum* and *Secale cereale*, respectively, with variable effects on host phenotype [74]. Nevertheless, it is noteworthy that although some of the materials inoculated with *Epichloë* showed successful performance, there are potential intergenerational transmission barriers due to the incompatibility between certain *Epichloë* species and specific hosts. Many successful cases have demonstrated the practicality of the *Epichloë* species in grass germplasm innovation. For instance, researchers utilized the well-characterized *Epichloë* strains NEA2, AR1, and AR37 to develop multiple commercially valuable grass cultivars, which comprised over 70% of proprietary seed sales in New Zealand a decade ago [75]. All isolates examined in this study provide the necessary materials for utilizing *Epichloë* for plant germplasm innovation. However, further experiments are needed to assess their ability to produce alkaloids in hosts, with the potential for application in artificial inoculation studies.

So far, there have been no reports of natural infection of cereal crops by the *Epichloë* species. Given the significance of cereal crops, there is an increasing interest in exploring the *Epichloë* species to accelerate the breeding of new cereal crop varieties with specific traits. The prerequisites for this are that the *Epichloë* provide an enhanced adaptability potential for their non-natural hosts [74,76], although obtaining and optimizing this benefit in these grains will require subsequent breeding or even genetic engineering to address the variability and compatibility of the *Epichloë* effect. Researchers have investigated the inoculation of *E. bromicola* strains isolated from wild barley to cultivated barley, which show a close genetic relatedness to wild barley. The report highlights the significant achievements, such as an increase in above-ground biomass, seed production per plant, and an earlier onset of the growing period [76]. The experimental operation involving fungal cultivation and inoculation of the *Epichloë* species indicates that the degree of genetic similarity between the natural and new host plants is positively correlated with the possibility of establishing a mutually symbiotic relationship [74]. It is imperative to explore *Epichloë* species in the wild relatives of cultivated plants. While challenges remain in optimizing *Epichloë* species inoculation and understanding the mechanisms underlying host compatibility, successful applications in cereal crops’ breeding underscore the utility of these endophytes in generating novel germplasm with desirable attributes. The *Epichloë* analyzed in this study were isolated from *Elymus* species (tribe Triticeae), which have a relatively close genetic relationship with barley, rye, and wheat, highlighting their potential in the genetic improvement of barley, rye, wheat, and their specific traits. The finding of ergovaline, paspaline, chanoclavine I, and D-lysergic produced by some *El. dahuricus* and *El. mutabilis* isolates illustrates the need to caution that these fungal toxins might also be introduced into crops through this novel, seed-transmitted symbiotic relationship.

Although alkaloids are generally regarded as the direct mediators of host defense, new evidence suggests that the *Epichloë* species may also activate the indirect defense pathways of grasses. Previous studies mainly focused on alkaloid-producing *Epichloë* spp. that simultaneously activate both direct and indirect defense mechanisms of host grasses, thereby complicating the unraveling of their mechanisms. In the current research, it was found that five isolates had no theoretical potential to synthesize any kinds of alkaloids; should these isolates be unable to synthesize alkaloids from all four main categories, it may serve as a valuable model for exploring the mechanism of how *Epichloë* species activate host defense and for underling alkaloid gene loss during evolution. This system can also provide a comprehensive framework for studying the *Epichloë*-mediated indirect plant defense mechanisms, thereby reducing the confounding factors in metabolite analysis.

## 5. Conclusions

In conclusion, *El. cylindricus* and *El. tangutorum* were identified as previously unreported hosts of *Epichloë* endophytes in China. Morphological examination and phylogenetic analyses confirmed the identity of 30 strains as *E*. *bromicola*. The alkaloid biosynthesis gene profiles in *El. cylindricus* and *El. tangutorum* were first investigated and it was found that the isolates only have the potential to produce some early pathway intermediates, and intermediates within the indole-diterpene alkaloids and ergot alkaloids pathway, such as paspaline, chanoclavine I, D-lysergic acid, and ergovaline, while no peramine and loline would be expected. Considering the toxicity of these isolates to the safety of herbivorous livestock, the five types of strains can be divided into two categories. Category I includes five animal-safe strains of type V, which do not produce any kinds of alkaloids, provide excellent basic materials for artificial inoculation in the future, and can be fully utilized in the resistance breeding of Poaceae plants. However, further experimental studies are required to evaluate the alkaloid-producing capabilities of these endophytes in cultivated hosts, with potential implications for applications in artificial inoculation programs. Category II includes the remaining 25 strains which could produce indole-diterpene (paspaline) and/or ergot alkaloids (chanoclavine I, D-lysergic acid, ergovaline) that are toxic to herbivorous livestock. This research advances the knowledge of *Epichloë* endophytes by identifying the potential for specific alkaloid production and documenting two new symbiotic relationships in China. Based on successful cases, it is advisable to continue exploring the diversity of *Epichloë* endophytes species and their diversity of alkaloid profiles to further investigate their interactions with different pasture species and their potential to enhance host stress resistance.

## Figures and Tables

**Figure 1 microorganisms-13-02275-f001:**
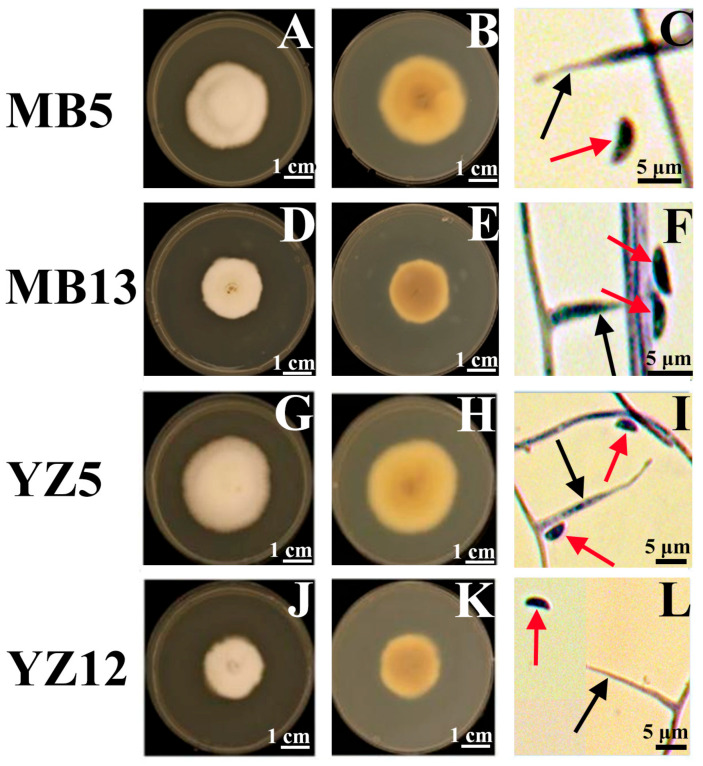
Colony morphology, conidia, and conidiogenous cells of selected *Epichloë* endophyte isolated from *Elymus tangutorum* and *Elymus cylindricus*. (The colony is from cultures grown on PDA at 25 °C for 32 d; MB5 and MB13 were isolated from *El. tangutorum*, YZ5 and YZ12 were isolated from *El. cylindricus*. The surface view of colonies (**A**,**D**,**G**,**J**). The reverse view of colonies (**B**,**E**,**H**,**K**). The micrographs of conidiogenous cells (black arrow) and conidia (red arrow) (**C**,**F**,**I**,**L**), black scale bars = 5 μm, white scale bars = 1 cm).

**Figure 2 microorganisms-13-02275-f002:**
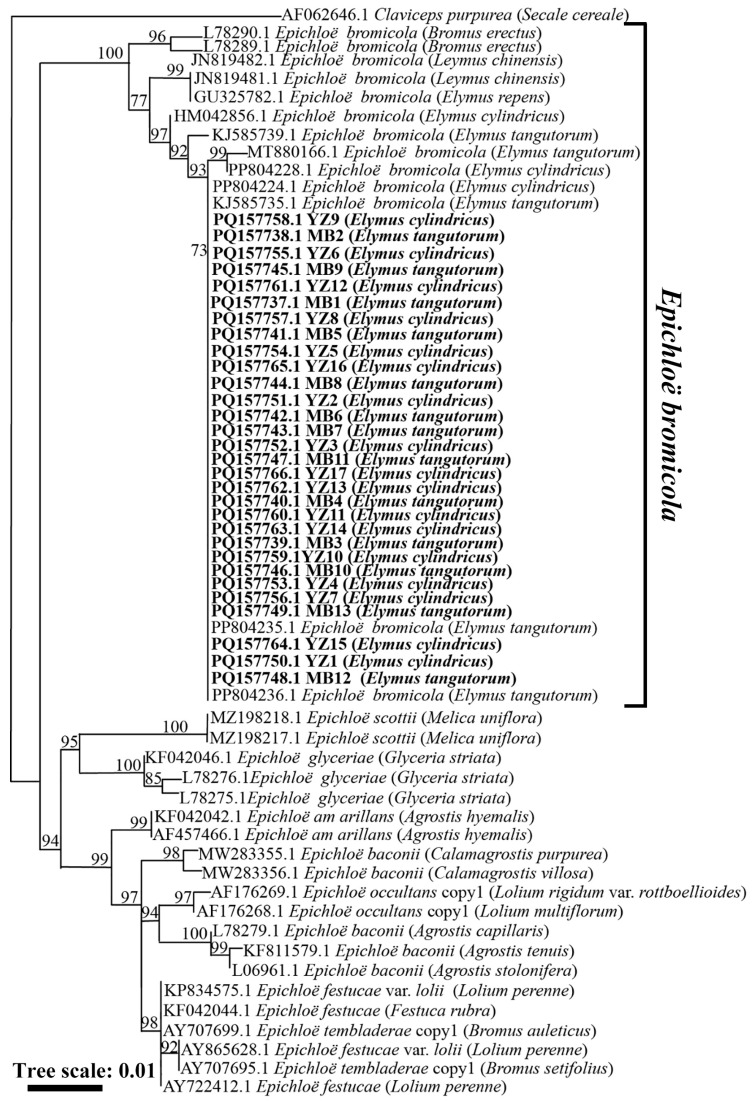
Phylogeny derived from a maximum likelihood (substitution model K2P + I) analysis of the *tubB* gene from representative *Epichloë* spp. The tree was rooted with *Claviceps purpurea* as the outgroup.

**Figure 3 microorganisms-13-02275-f003:**
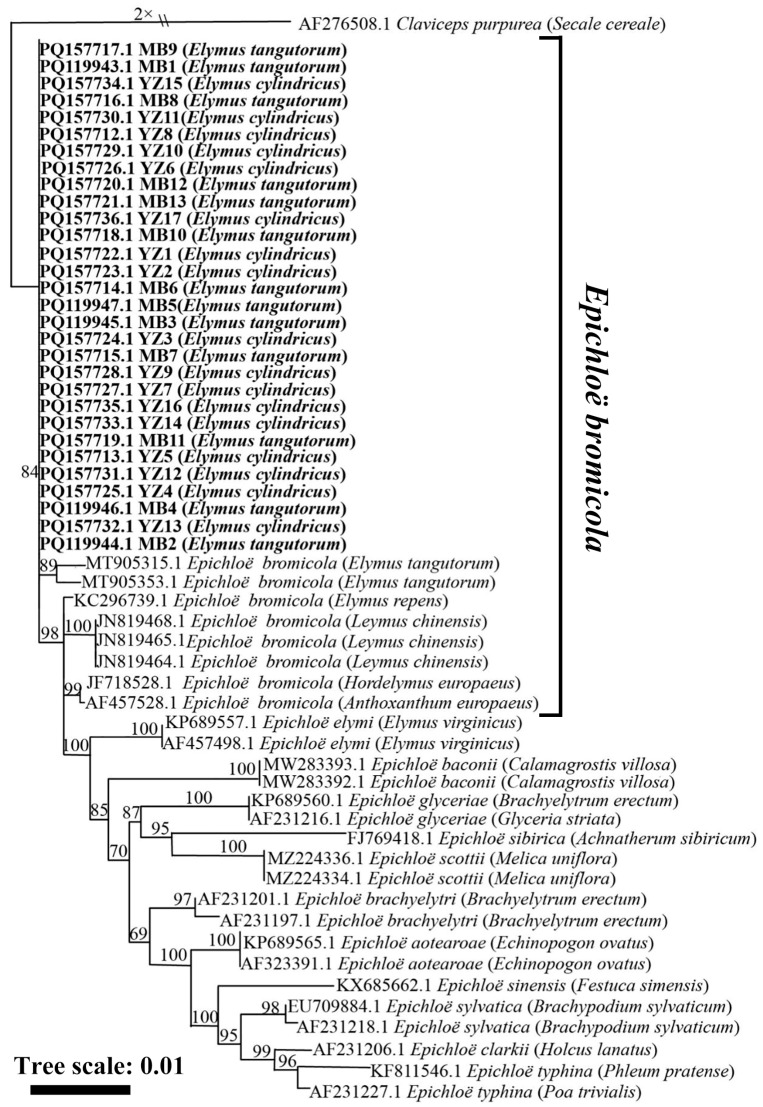
Phylogeny derived from a maximum likelihood (substitution model K2P + R2) analysis of the *tefA* gene from representative *Epichloë* spp. The tree was rooted with *Claviceps purpurea* as the outgroup.

**Table 1 microorganisms-13-02275-t001:** The seed lots of *Elymus cylindricus* and *Elymus tangutorum* used in this study.

Host	Location	Longitude	Latitude	Altitude (m)	No. of Samples	No. of Infected Samples	Infection Frequency (%)	No. of Isolated Strains
*Elymus cylindricus*	Qinghai, Haixi	98.5042° E	36.9185° N	2924.2	71	14	19.72	3
		98.5582° E	37.0036° N	3074.2	27	4	14.81	2
		97.3324° E	37.3628° N	2932.5	25	2	8.00	1
		97.2155° E	37.3197° N	2841.8	20	12	60.00	2
	Qinghai, Hainan	101.3243° E	35.8219° N	2906.3	60	10	16.67	2
		101.0984° E	35.7741° N	3268.1	30	6	20.00	2
		100.8079° E	34.7338° N	3344.6	30	2	6.67	1
	Qinghai, Guoluo	100.6322° E	34.6656° N	3212.1	30	4	13.33	1
		100.6789° E	33.0516° N	3574.8	32	4	12.50	1
	Qinghai, Yushu	97.0268° E	32.9638° N	3694.9	17	1	5.88	1
		96.4477° E	32.3355° N	3625.2	30	2	6.67	1
*Elymus tangutorum*	Qinghai, Haixi	97.2155° E	37.3197° N	2841.8	33	3	9.09	2
		97.7083° E	36.0147° N	3004.9	10	4	40.00	2
		97.8082° E	36.0430° N	2981.3	28	9	32.14	3
		98.1301° E	36.4181° N	3071.5	18	1	5.56	2
	Qinghai, Hainan	100.6789° E	33.0515° N	3329.7	24	3	12.50	2
		96.5569° E	32.6282° N	3574.8	6	1	16.67	1
	Qinghai, Yushu	97.0268° E	32.9638° N	3694.9	23	1	4.34	1

**Table 2 microorganisms-13-02275-t002:** Morphological characteristics of *Epichloë bromicola* isolated from *Elymus cylindricus*, *Elymus tangutorm* in this study, and *E. bromicola* endophytes isolated from *Elymus* spp. as references.

Host	Endophyte	Conidia Size (μm)		Length of Conidiogenous Cell (μm)	Growth Rate (mm/d, 25 °C)	Reference or Source
		Length	Width			
*Elymus cylindricus*	YZ1 (ns) ^1^	3.71 ± 0.12 (ef) ^1^	1.68 ± 0.17 (kl)	9.34 ± 0.38 (kl)	1.05 ± 0.02 (abcd)	This study
	YZ2 (ns)	4.64 ± 0.16 (b)	2.49 ± 0.16 (fgh)	10.47 ± 0.18 (j)	0.88 ± 0.08 (cdef)	
	YZ3 (ns)	4.58 ± 0.11 (bc)	2.56 ± 0.20 (efgh)	13.40 ± 0.24 (e)	1.01 ± 0.08 (abcd)	
	YZ4 (ns)	4.52 ± 0.19 (bc)	2.56 ± 0.19 (efgh)	14.37 ± 0.22 (d)	1.00 ± 0.09 (abcde)	
	YZ5 (ns)	4.60 ± 0.18 (bc)	2.53 ± 0.21 (efgh)	15.16 ± 0.22 (c)	1.03 ± 0.06 (abcd)	
	YZ6 (ns)	3.45 ± 0.13 (g)	1.51 ± 0.18 (m)	11.48 ± 0.25 (h)	1.18 ± 0.06 (a)	
	YZ7 (ns)	3.60 ± 0.13 (fg)	1.76 ± 0.14 (jk)	12.38 ± 0.43 (fg)	0.87 ± 0.23 (def)	
	YZ8 (ns)	3.49 ± 0.23 (g)	1.27 ± 0.04 (n)	9.09 ± 0.63 (l)	0.84 ± 0.05 (defg)	
	YZ9 (ns)	3.53 ± 0.16 (fg)	1.53 ± 0.04 (lm)	15.48 ± 0.26 (bc)	0.66 ± 0.11 (fgh)	
	YZ10 (ns)	3.53 ± 0.15 (fg)	1.86 ± 0.03 (j)	10.50 ± 0.22 (i)	0.67 ± 0.07 (fgh)	
	YZ11 (ns)	4.55 ± 0.17 (bc)	2.54 ± 0.09 (efgh)	9.38 ± 0.30 (kl)	0.64 ± 0.03 (gh)	
	YZ12 (ns)	4.39 ± 0.57 (c)	2.40 ± 0.13 (hi)	12.55 ± 0.29 (f)	0.68 ± 0.03 (fgh)	
	YZ13 (ns)	4.52 ± 0.20 (bc)	2.45 ± 0.07 (h)	12.69 ± 0.83 (f)	0.67 ± 0.03 (fgh)	
	YZ14 (ns)	4.63 ± 0.22 (b)	2.79 ± 0.07 (b)	13.37 ± 0.17 (e)	0.61 ± 0.04 (h)	
	YZ15 (ns)	4.51 ± 0.25 (bc)	2.51 ± 0.14 (efgh)	13.75 ± 0.28 (e)	0.90 ± 0.19 (bcde)	
	YZ16 (ns)	4.39 ± 0.17 (c)	2.60 ± 0.17 (defg)	12.59 ± 0.24 (f)	1.07 ± 0.07 (abcd)	
	YZ17 (ns)	3.59 ± 0.24 (fg)	2.29 ± 0.15 (i)	12.67 ± 0.23 (f)	0.63 ± 0.34 (gh)	
*Elymus tangutorm*	MB1 (ns)	5.42 ± 0.21 (a)	3.28 ± 0.15 (a)	16.55 ± 0.23 (a)	0.99 ± 0.35 (abcde)	This study
	MB2 (ns)	3.63 ± 0.23 (fg)	1.66 ± 0.24 (kl)	15.83 ± 0.60 (b)	0.91 ± 0.13 (bcde)	
	MB3 (ns)	3.50 ± 0.22 (fg)	1.79 ± 0.17 (jk)	15.47 ± 0.23 (bc)	1.06 ± 0.17 (abcd)	
	MB4 (ns)	5.47 ± 0.17 (a)	3.37 ± 0.25 (a)	10.91 ± 0.57 (h)	1.03 ± 0.09 (abcd)	
	MB5 (ns)	4.50 ± 0.15 (bc)	2.56 ± 0.19 (efgh)	12.07 ± 0.63 (g)	0.93 ± 0.12 (bcde)	
	MB6 (ns)	4.52 ± 0.20 (bc)	2.51 ± 0.10 (efgh)	11.61 ± 0.34 (h)	1.13 ± 0.04 (ab)	
	MB7 (ns)	4.61 ± 0.14 (bc)	2.43 ± 0.32 (hi)	9.56 ± 0.29 (k)	1.02 ± 0.14 (abcd)	
	MB8 (ns)	4.50 ± 0.09 (bc)	2.63 ± 0.15 (cdef)	12.47 ± 0.15 (f)	1.01 ± 0.13 (abcd)	
	MB9 (ns)	4.18 ± 0.35 (d)	2.66 ± 0.13 (bcde)	12.51 ± 0.21 (f)	1.01 ± 0.04 (abcd)	
	MB10 (ns)	5.34 ± 0.21 (a)	2.55 ± 0.16 (efgh)	10.49 ± 0.29 (j)	0.90 ± 0.06 (bcde)	
	MB11 (ns)	4.59 ± 0.16 (bc)	2.60 ± 0.17 (defg)	11.64 ± 0.74 (h)	1.02 ± 0.04 (abcd)	
	MB12 (ns)	4.58 ± 0.14 (bc)	2.76 ± 0.08 (bcd)	14.37 ± 0.72 (d)	1.11 ± 0.05 (abc)	
	MB13 (ns)	3.86 ± 0.14 (e)	2.77 ± 0.07 (bc)	13.64 ± 0.52 (e)	0.78 ± 0.21 (efg)	
*Bromus ramosus* (*Bromus benekenii*)	*Epichloë bromicola* (ns)	4.2 ± 0.5	2.0 ± 0.3	nt	0.90	[39]
*Bromus erectum*	*Epichloë bromicola* (s) ^1^	3.8 ± 0.4	2.0 ± 0.3	8–23	2.29–2.48 (PDA, 24 °C)	[39]
*Hordeum brevisubulatum*	*Epichloë bromicola* WBE1 (ns)	5.17 ± 0.06	2.87 ± 0.17	19.50 ± 1.06	0.88 ± 0.01 (PDA, 25 °C)	[42]
*Hordelymus europaeus*	*Epichloë bromicola* (ns)	4.2 ± 0.4	2.1 ± 0.2	20.2 ± 4.7	1.43–1.67 (PDA, 24 °C)	[43]
*Hordeum bogdanii*	*Epichloë bromicola* (ns)	4.6 ± 0.4	2.7 ± 0.3	14.0 ± 3.5	1.21 ± 0.1 (PDA, 25 °C)	[44]
*Leymus chinensis*	*Epichloë bromicola* (s)	5.3 ± 0.1	3.5 ± 0.1	29.0–31.0	1.7 ± 0.07 (PDA, 25 °C)	[45]
*Psathyrostachys lanuginosa*	*Epichloë bromicola* PF9 (ns)	3.6 ± 0.07	1.8 ± 0.04	12.3 ± 0.07	2.23 ± 0.05 (PDA, 25 °C)	[46]
*Roegneria kamoji*	*Epichloë bromicola* (s)	4.7–5.2	2.0–2.9	16.5–25.8	0.83–2.58 (PDA, 25 °C)	[47,48]
*Elymus cylindricus*	*Epichloë bromicola* AD3 (ns)	4.16 ± 0.07	nt	10.75 ± 0.76	1.01 ± 0.04 (PDA, 25 °C)	[36]
*Elymus dahuricus*	*Epichloë bromicola* LE1 (ns)	3.33 ± 0.09	nt	11.85 ± 0.70	2.32 ± 0.06 (PDA, 25 °C)	[36]
*Elymus dahuricus*	*Epichloë bromicola* (s)	4.5–6.5	2.5–4	9–27	0.78–1.05 (PDA, 25 °C)	[49]
*Elymus nutans*	*Epichloë bromicola* GA2 (ns)	3.75 ± 0.10	nt	10.76 ± 0.47	0.83 ± 0.03 (PDA, 25 °C)	[36]
*Elymus sibiricus*	*Epichloë bromicola* FC1 (ns)	3.78 ± 0.09	nt	13.59 ± 0.91	1.13 ± 0.03 (PDA, 25 °C)	[36]
*Elymus tangutorum*	*Epichloë bromicola* KE1 (ns)	5.09 ± 0.13	nt	13.33 ± 0.62	1.29 ± 0.09 (PDA, 25 °C)	[36]
*Elymus virginicus*	*Epichloë elymi* (s)	4.0 ± 0.4	2.2 ± 0.2	17.0 ± 3.0	1.95–2.85, (PDA, 24 °C)	[50]
*Elymus canadensis*	*Epichloë canadensis* (ns)	5.8–8	2.5–4.0	12.5–41.5	0.96–1.71, (PDA, 23 °C)	[51]
*Elymus nutans*	*Epichloë glyceriae* (ns)	3.8–6.2	2.2–2.8	31.0 ± 5.0	2.47–3, (PDA, 24 °C)	[50]

^1^ Note: Different letters indicate a significant difference between the 30 *Epichloë* endophytes strains (*p* < 0.05). ns: non-stromal strain; s: stroma forming strain; nt: not tested.

**Table 3 microorganisms-13-02275-t003:** Mating type and segments of peramine biosynthesis genes (*ppzA*) in the genome of *Epichloë*.

Endophyte Isolate	Mating Type Genes		Segments of *ppzA* Gene							
	*mtAC*	*mtBA*	A1	T1	C	A2	M	T2	R	ΔR
YZ8, MB8, MB11	+	-	-	-	-	-	-	-	-	-
YZ1, MB1	+	-	-	-	-	-	-	-	-	+
YZ11, YZ13, MB13	+	-	-	-	+	-	-	-	-	-
YZ9, YZ14	+	-	-	-	-	-	-	+	-	+
YZ12, MB12	+	-	-	+	-	-	-	-	-	+
YZ7, MB7, MB9	+	-	-	-	+	-	-	+	-	+
YZ2, YZ4, MB2, MB4	+	-	-	-	+	+	-	+	-	+
YZ10	+	-	+	-	-	-	-	+	+	+
YZ5, MB5	+	-	+	-	+	+	-	+	-	+
YZ15	+	-	-	-	+	+	-	+	+	+
MB10	+	-	+	-	+	-	-	+	+	+
MB6	+	-	+	+	+	+	+	+	-	+
YZ3, YZ6, YZ16, YZ17, MB3	+	-	+	+	+	+	+	+	+	+

Note: “+” indicates the presence of PCR amplification, while “-” indicates its absence.

**Table 4 microorganisms-13-02275-t004:** Ergot alkaloid biosynthesis genes (*EAS*) in the genome of *Epichloë* endophytes.

Endophye	Ergot Alkaloid (*EAS*) Genes														Predicted Ergot-Alkaloid-Producing Type
	*dmaW*	*easF*	*easC*	*easE*	*easD*	*easA*	*easG*	*cloA*	*lpsB*	*lpsA*	*easH*	*lpsC*	*easO*	*easP*	
YZ5	+	-	+	-	-	-	-	-	-	-	+	-	-	-	---
MB5	+	+	+	+	-	+	-	-	-	-	-	-	-	-	CC
YZ9	+	-	+	+	+	+	+	+	-	-	-	-	-	-	---
YZ13	+	+	+	+	+	-	-	+	-	+	+	-	-	-	CC
MB6	+	+	+	+	+	+	+	+	-	-	-	-	-	-	CC, D-LA
MB12	+	+	+	+	+	+	-	+	+	-	-	-	-	-	CC
MB13	+	+	+	+	+	+	-	+	-	+	-	-	-	-	CC
YZ8	+	-	+	+	+	+	+	+	+	-	+	-	-	-	---
YZ12	+	+	+	+	+	+	-	+	-	+	+	-	-	-	CC
YZ14	+	-	+	+	+	+	+	+	-	+	+	-	-	-	---
MB2	+	+	+	+	+	+	+	+	+	-	-	-	-	-	CC, D-LA
YZ7	+	-	+	+	+	+	+	+	+	+	+	-	-	-	---
YZ11	+	+	+	+	+	+	+	+	+	-	+	-	-	-	CC, D-LA
MB3, MB7, MB11	+	+	+	+	+	+	+	+	-	+	+	-	-	-	CC, D-LA
MB4	+	+	+	+	+	+	-	+	+	+	+	-	-	-	CC
MB8	+	+	+	+	+	+	+	+	+	+	-	-	-	-	CC, D-LA
YZ1, YZ2, YZ3, YZ4, YZ6, YZ10, YZ15, YZ16, YZ17, MB1, MB9, MB10	+	+	+	+	+	+	+	+	+	+	+	-	-	-	CC, D-LA, ERV

Note: “+” indicates the presence of PCR amplification, while “-” indicates its absence. CC = chanoclavine I; D-LA = D-lysergic acid; ERV = ergovaline; --- denotes limitations in synthesizing any type of ergot alkaloids.

**Table 5 microorganisms-13-02275-t005:** Indole-diterpene alkaloids biosynthesis genes (*IDT*/*LTM*) in the genome of *Epichloë* endophytes.

Endophyte	Indole-Diterpene (*IDT*/*LTM*) Genes											Predicted Indole-Diterpene-Producing Type
	*idtG*	*idtB*	*idtM*	*idtC*	*idtS*	*idtP*	*idtQ*	*idtF*	*idtK*	*idtE*	*idtJ*	
MB5, MB6, MB7	+	-	-	-	-	-	-	-	-	-	-	---
YZ9	-	+	-	-	+	-	-	-	-	-	-	---
YZ14	+	+	-	-	-	-	-	-	-	-	-	---
YZ1, YZ2	-	+	-	+	+	-	-	-	-	-	-	---
YZ8	+	+	-	-	-	-	-	+	-	-	-	---
MB1	-	+	-	-	+	-	-	+	-	-	-	---
MB9	+	+	-	-	-	-	-	+	-	-	-	---
MB12	-	+	-	-	+	-	-	+	-	-	-	---
MB13	+	-	-	+	-	-	-	+	-	-	-	---
YZ4	-	+	-	+	+	-	-	-	+	-	-	---
YZ7	+	+	-	+	+	-	-	-	-	-	-	---
YZ15	+	+	-	-	+	-	-	+	-	-	-	---
MB4	+	+	-	-	-	-	-	+	+	-	-	---
YZ3, YZ5	-	+	+	+	+	-	-	-	+	-	-	---
YZ13	+	+	-	+	-	-	+	+	-	-	-	---
MB2	+	-	-	+	+	-	-	+	+	-	-	---
YZ10, YZ11, YZ12	+	+	-	+	+	-	-	+	+	-	-	---
YZ6	+	+	+	+	-	-	+	+	+	-	-	---
MB3	-	+	+	+	+	-	+	+	+	-	-	---
MB8, MB11	+	+	-	+	+	-	+	+	+	-	-	---
YZ16, YZ17, MB10	+	+	+	+	+	-	+	+	+	-	-	paspaline

Note: “+” indicates the presence of PCR amplification, while “-” indicates its absence. --- denotes limitations in synthesizing any type of indole-diterpene alkaloids.

**Table 6 microorganisms-13-02275-t006:** The types of *Epichloë* endophytes isolated from *Elymus cylindricus* and *Elymus tangutorum*.

Type	Total No. of Strains	Mating Type Genes	Endophyte	*Epichloë* Species	Predicted Alkaloid-Producing Type
I	3	mtAC	YZ16, YZ17, MB10	*E. bromicola*	PAS + CC + D-LA + ERV
II	9	mtAC	YZ1, YZ2, YZ3, YZ4, YZ6, YZ10, YZ15, MB1, MB9	*E. bromicola*	CC + D-LA + ERV
III	7	mtAC	YZ11, MB2, MB3, MB6, MB7, MB8, MB11	*E. bromicola*	CC + D-LC
IV	6	mtAC	YZ12, YZ13, MB4, MB5, MB12, MB13	*E. bromicola*	CC
V	5	mtAC	YZ5, YZ7, YZ8, YZ9, YZ14	*E. bromicola*	---

Note: PAS = paspaline; CC = chanoclavine I; D-LA = D-lysergic acid; ERV: ergovaline; --- denotes limitations in synthesizing any type of alkaloids.

## Data Availability

The original contributions presented in this study are included in the article. Further inquiries can be directed to the corresponding author.

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
