# Peer review of "Morphological, Molecular, and Alkaloid Gene Profiling of Epichloë Endophytes in Elymus cylindricus and Elymus tangutorum from China"

_microorganisms, 2025, doi:10.3390/microorganisms13102275_

Round 1

Reviewer 1 Report

Comments and Suggestions for Authors

This study investigated Epichloë endophytes associated with Elymus cylindricus and Elymus tangutorum collected from the Qinghai–Tibet Plateau using morphological, molecular, and alkaloid gene profiling approaches. This topic is relevant to grassland ecology, livestock safety, and forage breeding. This study was well designed and provides new insights into intraspecific diversity in alkaloid biosynthetic genes. The manuscript has good potential but requires major revisions before it can be considered suitable for publication.

Abstract

  • L15–18: Please revise the grammar and clarity. Suggested: “Ergot alkaloids are toxic to invertebrates and mammals; peramine acts as an insect feeding deterrent; and loline alkaloids possess potent insecticidal activity.”
  • L20–24: Stating that all isolates were E. bromicola reduces the novelty of the study. Emphasize the intraspecific alkaloid gene diversity as the key novel outcome.
  • L25–29: Division into “safe” vs. “toxic” isolates is important. Please highlight the practical implications for livestock management and safe endophyte utilization in breeding.
  • Keywords (L34): Add “phylogenetics” and “mating type genes” to improve discoverability.

Introduction

  • L37–46: The life cycle description is too detailed; condense it to focus on the research gap.
  • L52–65: Expand the discussion on the toxicological risks of alkaloid-producing endophytes in Asian grazing systems, which strengthens the justification for this study.
  • L87–94: Update breeding-related references with recent work (e.g., studies post-2021 on safe endophytes in forage breeding).
  • L106–110: The claim that E. cylindricus and E. tangutorum endophytes are “not yet characterised” must be supported with stronger evidence and up-to-date references.

Materials and Methods

  • L119–127: Please provide the total number of plants sampled per location more clearly.
  • L129–137: Surface sterilization includes details of contamination control (e.g., plating of sterile water control).
  • L135–143: Please specify how the daily growth rate was calculated (time intervals, formula).
  • L139–146: Please clarify the microscopy method: magnification used, whether fluorescence or brightfield imaging.
  • L151–181: GenBank accession numbers for tefA and tubB are mentioned but missing in-text references to the corresponding isolates. Please explicitly link the isolate IDs with the accession numbers.

Results

  • L262–273: Phylogenetic analysis—bootstrap support for tefA was relatively weak (84). Please discuss why and whether more loci could strengthen the species delimitation.
  • L283–286: All isolates of mating type A are consistent with prior Elymus studies. Please compare more critically with reports of occasional MT-B isolates from other hosts.
  • Tables 3–6: The data presentation is comprehensive but difficult to follow. Consider graphical representation (heatmap or schematic pathway maps) to summarize the gene presence/absence.
  • Figures 1–3: The figures are informative but require improved legends (clearer scale bars, highlighting diagnostic traits).

Discussion

  • L350–357: Provide a stronger literature context by comparing infection frequency and choke disease occurrence in Elymus spp. globally.
  • L386–395: A good discussion on alkaloid diversity. Please integrate the findings from recent genomic studies.
  • L419–433: Ergovaline-producing strains pose a risk to livestock. Quantify potential exposure (e.g., % of grasslands dominated by these species and livestock grazing pressure).
  • L439–447: The breeding discussion is broad. Please provide concrete case studies of successful Epichloë inoculations in cereals or forages with outcomes.

Conclusion

  • The current conclusion repeats the discussion. Please shorten and emphasize the three key contributions.
    1. Confirmation of E. bromicola in E. cylindricus and E. tangutorum.
    2. Discovery of significant alkaloid biosynthetic diversity (five types).
    3. Implications for safe endophyte-based breeding and livestock-risk assessment.

Author Response

This study investigated Epichloë endophytes associated with Elymus cylindricus and Elymus tangutorum collected from the Qinghai–Tibet Plateau using morphological, molecular, and alkaloid gene profiling approaches. This topic is relevant to grassland ecology, livestock safety, and forage breeding. This study was well designed and provides new insights into intraspecific diversity in alkaloid biosynthetic genes. The manuscript has good potential but requires major revisions before it can be considered suitable for publication.

Response: Thanks, Sir, for your constructive comments. Thanks for your help in improving this manuscript.

Abstract

Questions # 1: L15–18: Please revise the grammar and clarity. Suggested: “Ergot alkaloids are toxic to invertebrates and mammals; peramine acts as an insect feeding deterrent; and loline alkaloids possess potent insecticidal activity.”

Response: Revised. Thanks, Sir, for your constructive comments. We have revised the lines according to your suggestions. Please see lines 17-19 (in track change version, same as below).

Questions # 2: L20–24: Stating that all isolates were E. bromicola reduces the novelty of the study. Emphasize the intraspecific alkaloid gene diversity as the key novel outcome.

Response: Revised. Thanks, Sir, for your constructive comments. We have revised the lines according to your suggestions. Please see lines 21-30.

Questions # 3: L25–29: Division into “safe” vs. “toxic” isolates is important. Please highlight the practical implications for livestock management and safe endophyte utilization in breeding.

Response: Revised. Thanks, Sir, for your constructive comments. We have divided 30 isolates into 2 categories, Category I includes 5 animal-safe strains, Category II includes the remaining 25 strains which could produce indole-diterpene (paspaline) and/or ergot alkaloids (chanoclavine I, D-lysergic acid, ergovaline) that are toxic to herbivorous livestock. Please see lines 24-28.

Questions # 4: Keywords (L34): Add “phylogenetics” and “mating type genes” to improve discoverability.

Response: Revised. Thanks, Sir. We have added phylogenetics and mating type genes, please see line 34.

Introduction

Questions # 5: L37–46: The life cycle description is too detailed; condense it to focus on the research gap.

Response: Revised. Thanks, Sir, for your constructive comments. We have revised the lines according to your suggestions. Please see lines 44-47. Species of the Epichloë genus that reproduce sexually are capable of forming fruiting bodies known as stromata, which envelop the developing inflorescences and inhibit seed formation—resulting in "choke" diseases in host plants, and consequently facilitate horizontal transmission.

Questions # 6: L52–65: Expand the discussion on the toxicological risks of alkaloid-producing endophytes in Asian grazing systems, which strengthens the justification for this study.

Response: Revised. Thanks, Sir, for your constructive comments. We have revised the lines according to your suggestions. Please see lines 115-117. The genes responsible for the biosynthesis of ergot alkaloids, indole-diterpenes, bispyrrolidines (loline) and pyrrolipyrazines (peramine) in Epichloë endophyte symbiotic with El. cylindricus and El. tangutorum from Qinghai-Tibet Plateau have not yet been characterised.

Questions # 7: L87–94: Update breeding-related references with recent work (e.g., studies post-2021 on safe endophytes in forage breeding).

Response: Revised. Thanks, Sir, for your constructive comments. We have added reference (Caradus, J.R.; Card, S.D.; Hewitt, K.G.; Hume, D.E.; Johnson, L.J. Asexual Epichloë fungi—obligate mutualists. Encyclopedia 2021, 6871, 1084-1100, doi:10.3390/encyclopedia1040083). Please see lines 63-75.

Questions # 8: L106–110: The claim that E. cylindricus and E. tangutorum endophytes are “not yet characterised” must be supported with stronger evidence and up-to-date references.

Response: Revised. Thanks, Sir, for your constructive comments. We have revised the sentence to “The genes responsible for the biosynthesis of ergot alkaloids, indole-diterpenes, bispyrrolidines (loline) and pyrrolipyrazines (peramine) in Epichloë endophyte symbiotic with El. cylindricus and El. tangutorum from Qinghai-Tibet Plateau have not yet been characterised” . Please see lines 115-117.

Materials and Methods

Questions # 9: L119–127: Please provide the total number of plants sampled per location more clearly.

Response: Revised. Thanks, Sir, for your constructive comments. We have added the results of the total number of plants sampled per location (No. of Samples) in Table 1. lines according to your suggestions. Please see lines 63-75

Questions # 10: L129–137: Surface sterilization includes details of contamination control (e.g., plating of sterile water control).

Response: Revised. Thanks, Sir, for your constructive comments. We have revised the lines according to your suggestions. Please see lines 141-142. and then rinsed three consecutive times with sterile water, and left onto sterile filter paper to dry.

Questions # 11: L135–143: Please specify how the daily growth rate was calculated (time intervals, formula).

Response: Revised. Thanks, Sir, for your constructive comments. We have revised the lines according to your suggestions. Please see lines 150-152. colony diam measured weekly over a 56-d period using the cross-crossing method, and the daily growth rate for each strain was calculated [37,40,41], The results presented are expressed as the mean ± standard error (Table 2).

Questions # 12: L139–146: Please clarify the microscopy method: magnification used, whether fluorescence or brightfield imaging.

Response: Revised. Thanks, Sir, for your constructive comments. We have revised the lines according to your suggestions. Please see lines 154-159. the morphology of conidia and conidiogenous cells was observed and photographed using an fluorescence microscope (Olympus BX63, Olympus, Tokyo, Japan). Images were captured with an Olympus BX51 camera supported by Cellsens Entry 1.8 software (Olympus Corp.). Images were used to measure the length and width of conidia (30 conidia were measured for each isolate) and the length of conidiogenous cells (20 conidiogenous cells were measured for each isolate).

Questions # 13: L151–181: GenBank accession numbers for tefA and tubB are mentioned but missing in-text references to the corresponding isolates. Please explicitly link the isolate IDs with the accession numbers.

Response: Thanks, Sir, for your remind. Accession numbers of each isolates were linked in Figure 2 and Figure 3. Please see Figures in the text.

Results

Questions # 14: L262–273: Phylogenetic analysis—bootstrap support for tefA was relatively weak (84). Please discuss why and whether more loci could strengthen the species delimitation.

Response: Thank you for the comment. The relatively weak bootstrap support for the main clade in the tefA tree (84) is possibly due to the limitations of single-locus data, including fewer informative sites or locus-specific conflict, which is a common issue in phylogenetics. However, the results from the tubB gene provided unequivocal support (bootstrap 100) for the same clade, and the agreement between the two loci increased our confidence in species delimitation. Multi-locus analyses are strongly recommended for robust phylogenetic inference and species boundaries. The congruent tree topology and combined data of tefA and tubB demonstrated that all 30 endophyte isolates from Elymus cylindricus and E. tangutorum are reliably placed within Epichloë bromicola.

Our findings are aligned with those of Linde et al. (2014), (Linde, C.C., Phillips, R.D., Crisp, M.D. and Peakall, R., 2014. Congruent species delineation of Tulasnella using multiple loci and methods. New Phytologist, 201(1), pp.6-12.) who reported that single-locus phylogenies, including tefA, often yielded weak bootstrap support hindering confident species delimitation. Through multilocus analyses, including tubB sequences, bootstrap support substantially improved, resulting in well-supported and congruent species clades. Similarly, our multilocus approach markedly strengthens the resolution of E.bromicola species boundaries in Elymus hosts, overcoming the limitations inherent in single-gene analyses.

Our observation of moderate bootstrap support from tefA, substantially increased by tubB, is consistent with the findings of Wang et al. (2021), (Wang, W., Hejasebazzi, A., Zheng, J. and Liu, K.J., 2021. Build a better bootstrap and the RAWR shall beat a random path to your door: phylogenetic support estimation revisited. Bioinformatics, 37(Supplement 1), pp.i111-i119.) who reported that single-locus bootstrap analyses often show weak or variable support. Their multilocus bootstrap method (RAWR) highlights the improved confidence achieved via combining loci. This provides robust evidence for species boundaries, supporting our identification of the E. bromicola clade despite limited resolution from tefA alone.

Questions # 15: L283–286: All isolates of mating type A are consistent with prior Elymus studies. Please compare more critically with reports of occasional MT-B isolates from other hosts.

Response: Thanks, Sir, for your constructive comments. We have added the comparasion in the discussion part. Please see lines 392-418.

Questions # 16: Tables 3–6: The data presentation is comprehensive but difficult to follow. Consider graphical representation (heatmap or schematic pathway maps) to summarize the gene presence/absence.

Response: Great Questions. Thanks for your constructive suggestion. Since the alkaloid metabolism pathway is highly complex, involving 30 microbial strains. Given the constraints of space, presenting the data in tabular format provides the clearest and most concise representation. We have learned many related paper involving alkaloid genes, Similar situations, the authors all used tables to show their gene presence/absence, such as: 1. ‘Schardl, C. L., S. Florea, P. Nagabhyru, J. Pan, M. L. Farman, C. A. Young, M. Rahnama, A. Leuchtmann, M. R. Sabzalian, M. Torkian, A. Mirlohi and L. J. Iannone. Chemotypic diversity of bioprotective grass endophytes based on genome analyses, with new insights from a Mediterranean-climate region in Isfahan Province, Iran." Mycologia: 1-26.’; 2. ‘Thunen, T., Y. Becker, M. P. Cox and S. Ashrafi (2022). "Epichloe scottii sp. nov., a new endophyte isolated from Melica uniflora is the missing ancestor of Epichloe disjuncta." IMA Fungus 13(1): 2.’; 3. ‘Tian, P., W. Xu, C. Li, H. Song, M. Wang, C. L. Schardl and Z. Nan (2020). Phylogenetic relationship and taxonomy of a hybrid Epichloë species symbiotic with Festuca sinensis." Mycological Progress 19(10): 1069-1081.’; 4. ‘Charlton, N. D., K. D. Craven, S. Mittal, A. A. Hopkins and C. A. Young (2012). "Epichloe canadensis, a new interspecific epichloid hybrid symbiotic with Canada wildrye (Elymus canadensis)." Mycologia 104(5): 1187.’. Thanks for your constructive suggestion. In future papers, we will adopt your constructive suggestions.

Questions # 17: Figures 1–3: The figures are informative but require improved legends (clearer scale bars, highlighting diagnostic traits).

Response: Revised. Thanks for your constructive comments. We have improved figures, improved the scale bars to make it more clear and highlighting the diagnostic traits.

Discussion

Questions # 18: L350–357: Provide a stronger literature context by comparing infection frequency and choke disease occurrence in Elymus spp. globally.

Response: Revised. Thanks, Sir, for your constructive comments. We have revised the lines according to your suggestions. Please see lines 407-417. “Surveys of Epichloë endophyte infection in El. dahuricus across its natural distribution range in Xinjiang, Shanxi, and Beijing, China, have revealed highly variable infection frequencies, ranging from 0% to 100%. E. bromicola is the predominant, and often sole, Epichloë species associated with El. dahuricus; however, its sexual stage has not been observed in this host [59]. Similarly, the sexual reproductive stage of E. bromicola strains isolated from five Elymus species collected in various regions of Northwest China remains unobserved [36]. Originally identified from Bromus spp. grasses, E. bromicola was first described as a common choke disease pathogen of B. erectus and as a strictly seed-transmitted endophyte in B. benekenii and B. ramosus within the grass tribe Bromeae [36]. The formation of stromata in E. cylindricus and E. tangutorum warrants further investigation”.

Questions # 19: L386–395: A good discussion on alkaloid diversity. Please integrate the findings from recent genomic studies.

Response: Revised. Thanks, Sir, for your constructive comments. We have revised the lines according to your suggestions. Please see lines 434-497.

Questions # 20: L419–433: Ergovaline-producing strains pose a risk to livestock. Quantify potential exposure (e.g., % of grasslands dominated by these species and livestock grazing pressure).

Response: Revised. Thanks, Sir, for your constructive comments. We have revised the lines according to your suggestions. Please see lines 472-482. Although the risk of toxicosis may be much less in rangelands of China due to relative high diversity of forage and other grasses, the significance of ergovaline, ergonovine and ergine production by Epichloë of important forage grasses such as El. cylindricus and El. tangutorum warrants consideration for livestock that graze them and livestock management in rangelands of China. No livestock poisoning incidents were observed in the corresponding collection area of this study. This is primarily attributed to the insufficient concentration of toxic alkaloids to exceed the threshold required for livestock toxicity, or to the low biological activity of these alkaloids, which is inadequate to induce poisoning. However, further experimental investigation is necessary to assess their alkaloid production capacity in host grasses, with potential implications for applications in artificial inoculation studies.

Questions # 21: L439–447: The breeding discussion is broad. Please provide concrete case studies of successful Epichloë inoculations in cereals or forages with outcomes.

Response: Revised. Thanks, Sir, for your constructive comments. We have revised the lines according to your suggestions. Please see lines 100-106: “Through this procecess, a number of novel Epichloë isolates have been delivered and are now commercially used in USA and New Zealand, such as AR1TM, AR37TM, Endo5TM and NEATM endophyte for ryegrass, E34TM, AR542TM, MaxQTM, and MaxPTM for tall fescue. Effective delivery of novel Epichloë infected cultivars requires care with management of seed, quality control systems and monitoring of Epichloë viability is required through the distribution chain, the seed must be stored at relative low humidity and low temperature until ready to be sown”. And please see lines 496-598: “For instance, researchers utilized the well-characterized Epichloë strains NEA2, AR1 and AR37 to develop multiple commercially valuable grass cultivars, which comprised over 70% of proprietary seed sales in New Zealand a decade ago [74]”.

Conclusion

Questions # 22: The current conclusion repeats the discussion. Please shorten and emphasize the three key contributions.

Confirmation of E. bromicola in E. cylindricus and E. tangutorum.

Discovery of significant alkaloid biosynthetic diversity (five types).

Implications for safe endophyte-based breeding and livestock-risk assessment.

Response: Revised. Thanks, Sir, for your constructive suggestions. We have revised the lines according to your suggestions. Please see lines 536-558, we have deleted “We are currently assessing whether this diversity of alkaloids translates into differences in the adaptability and persistence of the hosts, and exploring the distribution of chemical diversity within the Elymus populations”. And we have added “Considering the toxicity of these isolates to the safety of herbivorous livestock, the 5 types of strains can be divided into 2 categories. Category I includes 5 animal-safe strains of type V, which do not produce any kinds of alkaloids, provided excellent basic materials for artificial inoculation in the future and can be fully utilized in the resistance breeding of Poaceae plants. However, further experimental studies are required to evaluate the alkaloid-producing capabilities of these endophytes in cultivated hosts, with potential implications for applications in artificial inoculation programs. Category II includes the remaining 25 strains which could produce indole-diterpene (paspaline) and/or ergot alkaloids (chanoclavine I, D-lysergic acid, ergovaline) that are toxic to herbivorous livestock.”.

Reviewer 2 Report

Comments and Suggestions for Authors

The majority of the comments are enclosed in the attached revised pdf. The authors must eliminate all the personalization and clarify the manuscript is several point, moving also some of the sentences to the appropriate position in the manuscript. A part from the relevant need of improving the English also the fact that the work was carried out just on seeds not on host plants content must be clarified. The production of the diverse molecules listed should have been proved by growing and testing the plants to verify the content of dangerous molecules at least. A clear statment descibing this must be added in the discussion and conclusions. The work is just teorethical at this stage and this must be clearly stated.

Comments on the Quality of English Language

English need a severe revision also for clarifying several parts of the work done.

Author Response

The majority of the comments are enclosed in the attached revised pdf. The authors must eliminate all the personalization and clarify the manuscript is several point, moving also some of the sentences to the appropriate position in the manuscript. A part from the relevant need of improving the English also the fact that the work was carried out just on seeds not on host plants content must be clarified. The production of the diverse molecules listed should have been proved by growing and testing the plants to verify the content of dangerous molecules at least. A clear statment descibing this must be added in the discussion and conclusions. The work is just teorethical at this stage and this must be clearly stated.

Response: Thanks, Sir, for your constructive comments. Thanks for your help in improving this manuscript. We wish to express our sincere gratitude to you for the valuable time and effort spent reviewing our manuscript and for providing constructive and insightful comments. We have found these suggestions to be extremely helpful in strengthening the quality and clarity of our work. Based on your comments, we have made the corrections to make the results more clear. Some of the sentences in results section were moved to the discussion part or conclusion part, please see lines 543-552 (in track change version, same as below). we also added some sentence to state that our work is just thorethical at this stage. such as we have added sentence “No livestock poisoning incidents were observed in the corresponding collection area of this study. This is primarily attributed to the insufficient concentration of toxic alkaloids to exceed the threshold required for livestock toxicity, or to the low biological activity of these alkaloids, which is inadequate to induce poisoning. However, further experimental investigation is necessary to assess their alkaloid production capacity in host grasses, with potential implications for applications in artificial inoculation studies. ” in lines 472-477. We invited Professor James White (an expert working on endophytes from USA) who is a native English speaker, has improved the language of our manuscript.  we have added “However, further experimental studies are required to evaluate the alkaloid-producing capabilities of these endophytes in cultivated hosts,” in conclusion part, please see lines 547-549. The specific modifications are as follows:

In line 15: We have revised “Epichloë endophytes are mutualistic associates of grasses, conferring benefits such as competitiveness, increased stress tolerance, and ecological dominance to host plants” to “Epichloë endophytes are mutualistic associates with grasses, conferring host plants with enhanced competitiveness, improved stress tolerance, and increased ecological dominance”.

In line 34: We have revised “Epichloë endophyte; Elymus cylindricus; Elymus tangutorum; alkaloid diversity” to “phylogenetics; mating type genes; alkaloid diversity”.

In line 40: We have deleted “, hosts are served as the habitants for the Epichloë”.

In lines 44-47: We have improved “The sexually reproducing species of the Epichloë genus can form fruiting bodies (stromata) that engulf the develop-ing inflorescences and inhibit seed formation (causing "choke" diseases) of hosts, thereby resulting in horizontal transmission” to “Species of the Epichloë genus that reproduce sexually are capable of forming fruiting bodies known as stromata, which envelop the developing inflorescences and inhibit seed formation—resulting in "choke" diseases in host plants, and consequently facilitate horizontal transmission. ”.

In lines 119-121: We have improved “The Elymus tangutorm and Elymus cylindrical were collected from August 2023 to September 2023, with mature seeds from Haixi, Hainan, Guoluo, and Yushu prefecture in Qinghai Province, China” to “The Elymus tangutorm and Elymus cylindrical specimens were collected between August and September 2023, with mature seeds obtained from Haixi, Hainan, Guoluo, and Yushu prefectures in Qinghai Province, China”.

In lines 140: We have deleted “for a while”.

In lines 143 and 147-148: We have added “Table 2”.

In line 148: We have revised to “The morphologies of the isolates were compared with those of other Epichloë species, including species isolated from Elymus spp.”.

In lines 158-161: We have deleted the base sequence of the primers.

In line 169: We have added “DNA fragments were stained with Gold View (Solarbio Corp.) and viewed by UV transillumination.”.

In lines 172-173: We have added “submitted to GenBank to confirm their belong-ing to the genus Epichloë and”.

In lines 185-189: We have improved to “Two mating-type genes, 8 segments of the ppzA gene, which is involved in peramine biosynthesis, 14 EAS cluster genes involved in ergot alkaloid biosynthesis, 11 IDT/LTM cluster genes required for indole-diterpene production, and 11 LOL cluster genes required for loline biosynthesis were identified.”.

In lines 190-196: We have added relevant references.

In lines 206-211: We have improved to “The Epichloë infection rate among plants from different regions ranged from 5.88% and 60%, with the highest rate was found in Haixi prefecture. For El. tangutorum, a total of 142 individual plants were collected from Haixi, Hainan, and Yushu prefectures, of which only 22 were found to carry Epichloë endophyte, yielding a total infection rate of 15.5%. The Epichloë infection rate for this species ranged between 4.34% and 40%, with Haixi prefecture again exhibiting the highest rate.”.

In lines 239-250: We have italicized the relevant Latin names.

In table 2: We have changed reference numbers, °C accordingly.

In lines 272-273: We have deleted “single-copy” and “and classified”.

In lines 284-285: We have improved to “During the amplification of mating type genes, 30 isolates in this study exhibited identical amplification profiles, being positive exclusively for the mtAC marker, which is indicative of mating type A (Table 3). ”.

In line 286: We have revised “formly perA gene” to “formly known as the perA gene”.

In lines 288-289: We have improved to “The ppzA-∆R (representing allele ppzA-2) refers to the ppzA gene from which the R-domain has been deleted, the functional implication of this deletion is the absence of the final enzymatic step required to convert diketopiperazine into peramine in the ∆R variant, resulting in the production of pyrrolopyrazine-1,4-diones instead of peramine.”.

In lines 293-296: We have improved to “Among the strains tested, 24 were positive for ppzA-∆R, suggesting their inability to produce peramine. In contrast, strains YZ8, MB8, MB11, YZ11, YZ13 and MB13 were negative for ppzA-∆R, but likely incapable of synthesizing peramine due to the absence of key functional domains within the ppzA gene.”.

In lines 299-306: We have improved to “Among the 14 genes involved in ergot alkaloid synthesis, 25 out of 30 strains were found to harbor the genes dmaW, easF, easC and easE, indicating their potential for synthesizing chanolavine I (CC). Seven strains MB2, MB3, MB6, MB7, MB8, MB11, and YZ11 contained 8 of the 14 genes associated with ergot alkaloids biosynthesis, including dmaW, easF, easC, easE, easD, easA, easG, and cloA, suggesting their potential to produce chanolavine I (CC) and D-lysergic acid (D-LA). Furthermore, 11 out of 14 genes at the EAS locus were identified in 11 strains, implying that these strains may be capable of synthesizing chanolavine I (CC), D-lysergic acid (D-LA) and ergovaline (ERV) , but no ergonovine and lysergic acid, based on the current understanding of the biosynthetic pathway. ”.

In line 309: We have revised “could not produce” to “may not produce”.

In lines 340-345: We have moved this part to Conclusions section. Please see lines 542-551.

In lines 356-357: We have improved to “5 isolates out of 30 were found to lack genes responsible for the biosynthesis of any alkaloids.”.

In lines 361-368: We have improved to “In a study by Du et al., (2024) , all 20 E. bromicola strains associated with five Elymus spp. from five regions in Northwest China were identified as mating type A (MTA) [36]. Yi et al., (2018) conducted a study analyzing E. bromicola from six Hordeum seed accessions, all of which were classified as MTA [61]. E. bromicola isolates obtained from symbiotic associations with H. brevisubulatum were also found to belong exclusively to MTA [57]. In another study, E. bromicola isolates derived from El. dahuricus revealed that all but one of the ten strains belonged to mating type A (MTA) [62]. Among eight E. bromicola strains isolated from El. kamoji, six were classified as mating type B (MTB), while the remaining two were categorized as MTA [63].”.

In lines 371-375: We have improved to “However, the simultaneous presence of both mating type A (MTA) and mating type B (MTB) in certain E. bromicola population suggests the potential existence of a sexual stage in specific cases, despite the absence of observed stromata on El. cylindricus and El. tangutorum under natural field conditions.”.

In lines 383-385: We have improved to “In addition, three strains isolated from surface-sterilized seeds of Hordeum bogdanii were observed all conformed to the  characteristics of morphological difference [66].”

In lines 410-412: We have improved to “Genome sequencing of two isolated Epichloë endophytes, F11 and E2368, from E. festucae revealed that strain F11 was capable of synthesizing the ergot alkaloid end products ergovaline and peramine, as well as the indole-diterpene end product lolitrem B. In contrast, E2368 produced only ergot alkaloid end products [68]. ”.

In lines 448-451: We have improved to “For instance, researchers utilized the well-characterized Epichloë strains NEA2, AR1 and AR37 to develop multiple commercially valuable grass cultivars, which comprised over 70% of proprietary seed sales in New Zealand a decade ago [77].”.

In lines 454-455: We have deleted “This will be the main focus of our future research.”.

In conclusion section: We have deleted “Epichloë bromicola has a wide range of hosts within the Poaceae family, and there are significant differences in the potential for alkaloid synthesis among different host species.” and “We are currently assessing whether this diversity of alkaloids translates into differences in the adaptability and persistence of the hosts, and exploring the distribution of chemical diversity within the Elymus populations.”. We further improved the discovery of significant alkaloid biosynthetic diversity (five types), please see lines 542-551.

Round 2

Reviewer 1 Report

Comments and Suggestions for Authors

The revised manuscript has improved considerably in both scientific clarity and structure. I appreciate the substantial effort made to address the reviewers’ earlier concerns, including the clearer abstract, the condensed life cycle section, the addition of up-to-date references, and the explicit acknowledgement that this work is theoretical at this stage. The separation of isolates into “safe” versus “toxic” categories and the incorporation of case studies from commercial breeding programs have enhanced the practical relevance of the study.

The methodology is now well described, with clearer details on sample numbers, growth rate calculations, microscopy, and GenBank accession linkage. The results are generally presented with adequate clarity, and the phylogenetic analyses are supported by appropriate discussion of bootstrap values and multilocus considerations. The expanded discussion on infection frequency, choke disease, alkaloid diversity, and livestock risk provides a stronger global context for the findings.

Nevertheless, there remain areas where the presentation could be improved:

  1. Tables 3–6 are very dense and difficult to interpret at a glance. Although I understand the justification provided, the inclusion of schematic summaries or heatmaps (even as supplementary figures) would significantly improve readability and accessibility for readers.

  2. Figures and legends have been improved but still require additional clarity—especially scale bars and diagnostic highlights for morphology-based figures.

  3. English expression has improved but could still benefit from careful polishing to eliminate awkward phrasing and grammatical inconsistencies. This would help the manuscript read more fluently and professionally.

Overall, the revised version is scientifically sound, novel, and provides meaningful insights into Epichloë–Elymus associations and their implications for grassland ecology, breeding, and livestock safety. With minor improvements in presentation and language, this study will make a valuable contribution to the field.

Reviewer 2 Report

Comments and Suggestions for Authors

Still a number of small revisions must be done see the list below:

page 5 lane 175: add city and country for the reagent

page 5 lane 212 erase was

page 7 lane 258 revise 25 °C to 25°C

page 9  in the whole table 2 erase the space between the temperature number and °C

page 10 lane 282 change formly to formerly

page 13 lanes 312, 314, 318, 321 replace the word strains with isolates

page 15 lane 350 replace Chanoclavinel with chanoclavinel

page 16 lane 381 replace strains with isolates

page 16 lane 382 replace strains isolated with isolates

page 17 lane 408 replace strains isolated with isolates

page 17 lane 409 rephrase to allow understanding of the meaning

page 17 lanes 413 and 433 replace strains with isolates

page 17 lane 417 replace its with their

page 17 lane 429 revise were unable with were probably unable

page 19 lane 535 replace our with the
